

# Evidence for a novel cranial thermoregulatory pathway in thalattosuchian crocodylomorphs

Mark T. Young[1,2], Charlotte I. W. Bowman[1], Arthur Erb[1], Julia A. Schwab[1,3], Lawrence M. Witmer[4], Yanina Herrera[5] and Stephen L. Brusatte[1]

[1] School of GeoSciences, University of Edinburgh, Edinburgh, United Kingdom
[2] LWL-Museum für Naturkunde, Münster, Germany
[3] Department of Earth and Environmental Sciences, University of Manchester, Manchester, United Kingdom
[4] Department of Biomedical Sciences, Ohio University, Athens, Ohio, United States
[5] Museo de La Plata, Facultad de Ciencias Naturales y Museo, Universidad Nacional de La Plata, La Plata, Argentina

Corresponding author
Mark T. Young,
marktyoung1984@gmail.com

## ABSTRACT

Thalattosuchian crocodylomorphs were a diverse clade that lived from the Early Jurassic to the Early Cretaceous. The subclade Metriorhynchoidea underwent a remarkable transition, evolving from semi-aquatic ambush predators into fully aquatic forms living in the open oceans. Thalattosuchians share a peculiar palatal morphology with semi-aquatic and aquatic fossil cetaceans: paired anteroposteriorly aligned grooves along the palatal surface of the bony secondary palate. In extant cetaceans, these grooves are continuous with the greater palatine artery foramina, arteries that supply their oral thermoregulatory structures. Herein, we investigate the origins of thalattosuchian palatal grooves by examining CT scans of six thalattosuchian species (one teleosauroid, two early-diverging metriorhynchoids and three metriorhynchids), and CT scans of eleven extant crocodylian species. All thalattosuchians had paired osseous canals, enclosed by the palatines, that connect the nasal cavity to the oral cavity. These osseous canals open into the oral cavity *via* foramina at the posterior terminus of the palatal grooves. Extant crocodylians lack both the external grooves and the internal canals. We posit that in thalattosuchians these novel palatal canals transmitted hypertrophied medial nasal vessels (artery and vein), creating a novel heat exchange pathway connecting the palatal vascular plexus to the endocranial region. Given the general hypertrophy of thalattosuchian cephalic vasculature, and their increased blood flow and volume, thalattosuchians would have required a more extensive suite of thermoregulatory pathways to maintain stable temperatures for their neurosensory tissues.

## INTRODUCTION

Thalattosuchian crocodylomorphs underwent a major evolutionary transition during the Jurassic, evolving from semi-aquatic nearshore predators to fully aquatic forms which lived

in the open oceans (*Fraas, 1902*; *Andrews, 1913*; *Buffetaut, 1982*; *Young et al., 2010*; *Wilberg, 2015*; *Ősi et al., 2018*; *Schwab et al., 2020*). Thalattosuchia is composed of two subgroups: Teleosauroidea, which evolved a diverse range of semi-aquatic morphologies but never made the transition to being fully aquatic (*Buffetaut, 1982*; *Foffa et al., 2019*; *Johnson, Young & Brusatte, 2020*); and Metriorhynchoidea, where the transition to life in the open ocean did occur (*Fraas, 1902*; *Young et al., 2010*; *Wilberg, 2015*; *Ősi et al., 2018*). Within Metriorhynchoidea, the fully aquatic subgroup Metriorhynchidae evolved a wide range of pelagic adaptations, including hydrofoil-like forelimbs, a hypocercal tail, loss of bony armour (osteoderms), and an osteoporotic-like lightening of the skull, femora and ribs (*e.g.*, *Fraas, 1902*; *Andrews, 1913*; *Hua & de Buffrénil, 1996*; *Young et al., 2010*). Metriorhynchids are also known to have had hypertrophied salt exocrine glands (*Fernández & Gasparini, 2000*, *2008*; *Fernández & Herrera, 2009*; *Herrera, Fernández & Gasparini, 2013*; *Cowgill et al., 2022a*) and smooth scaleless skin (*Spindler et al., 2021*). They possibly also evolved viviparity (see *Young et al., 2010*; *Herrera et al., 2017*) and an elevated metabolism (*Séon et al., 2020*).

Recently, computed tomography (CT) has been used to analyse the internal anatomy of thalattosuchian skulls, investigating their brains, sinuses, vasculature, salt glands and bony labyrinths (see *Fernández & Herrera, 2009*; *Fernández et al., 2011*; *Herrera, Fernández & Gasparini, 2013*; *Herrera, Leardi & Fernández, 2018*; *Brusatte et al., 2016*; *Pierce, Williams & Benson, 2017*; *Schwab et al., 2020*, *2021*; *Bowman et al., 2022*; *Cowgill et al., 2022a*, *2022b*; *Wilberg et al., 2022*). Thus, we are now beginning to get an unparalleled insight into the neurosensory and internal rostral soft-tissue anatomy of thalattosuchians, as well as the extensive changes that occurred within their crania as this group transitioned from being semi-aquatic to being fully aquatic.

One rostral structure that has not been investigated are the palatal grooves (sometimes also referred to as maxillo-palatine grooves, palatal canals, or anteroposterior sulci). All known metriorhynchoids have paired anteroposteriorly aligned grooves on the roof of the oral cavity, present on the palatal surface of the palatines and maxillae (Fig. 1; *Andrews, 1913*; *Parrilla-Bel et al., 2013*; *Foffa & Young, 2014*; *Aiglstorfer, Havlik & Herrera, 2020*; *Young et al., 2020a*, *2021*). While the grooves are clearly present in Early Jurassic teleosauroids (*Johnson et al., 2019*; *Johnson, Young & Brusatte, 2020*), in Middle and Late Jurassic taxa the grooves became progressively shallower and are perhaps absent in some genera of teleosaurids and machimosaurids (*e.g.*, *Johnson et al., 2018*; *Johnson, Young & Brusatte, 2020*; *Foffa et al., 2019*).

What formed the palatal grooves is unknown. Given that this feature is ubiquitous within Thalattosuchia (or at least in Early Jurassic thalattosuchians), but absent in other crocodylomorphs, it is possible that these grooves are linked to the land to sea transition that thalattosuchians underwent. To determine whether this is correct, here we investigate the palatal grooves in CT scans of six thalattosuchian species. We discovered that the posterior terminus of the grooves (on the palatines) is continuous with ossified canals that connect the oral cavity to the nasal cavity. Given their location, we hypothesise that these canals primarily held vasculature, and that the medial nasal arteries and veins, which are present in virtually all extant diapsids (*Porter & Witmer, 2015*, *2016*; *Porter, Sedlmayr &*
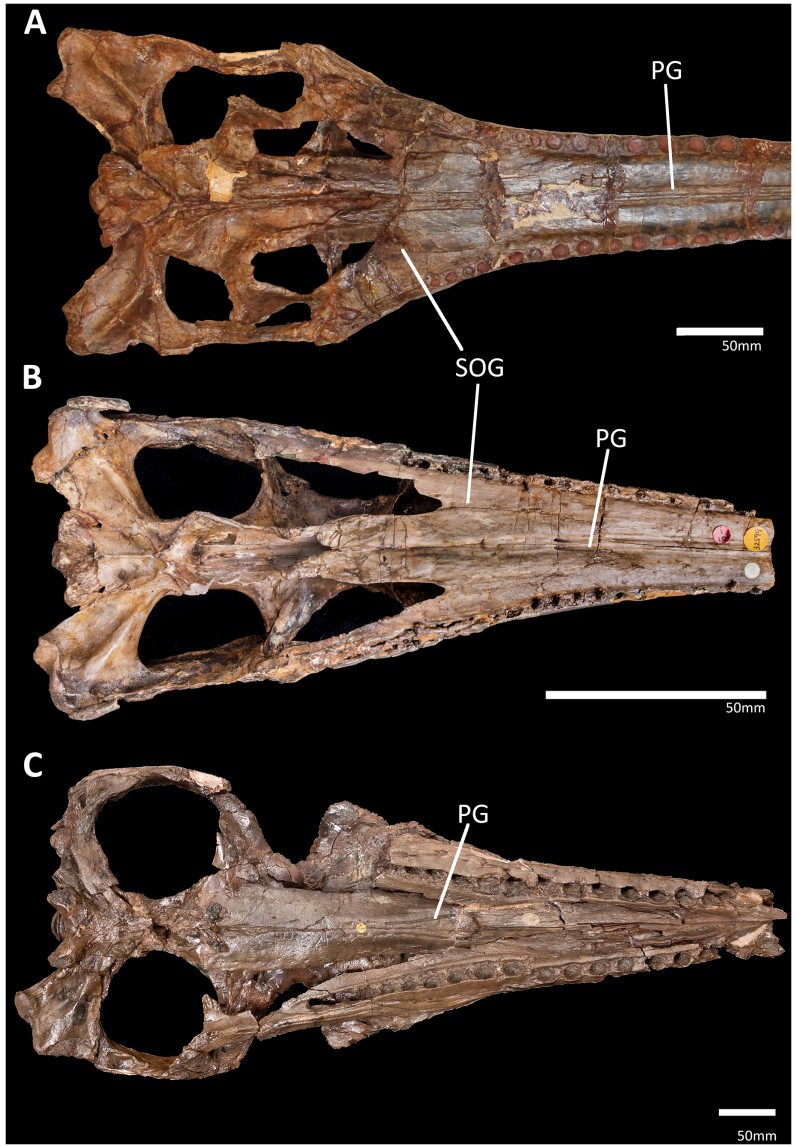

**Figure 1 Comparison of the palatal grooves in different thalattosuchian clades, skulls shown in palatal view.** (A) IVPP V 10098, Chinese teleosaurid; (B) NHMUK PV OR 32599, the early-diverging metriorhynchoid *Pelagosaurus typus*; (C) NHMUK PV R 3700, the metriorhynchid '*Metriorhynchus*' *brachyrhynchus*. Abbreviations: PG, palatal groove; SOG, suborbital groove.

*Witmer, 2016*), took a novel course and were transmitted along the external grooves. This would have connected the palatal vascular plexus to the ethmoid vessels, creating a new heat exchange pathway that would have helped moderate brain and eye temperatures.

## MATERIALS AND METHODS

We made internal rostral reconstructions of six thalattosuchian skulls based on CT scans (see Table S1). Our sample includes the teleosauroid *Plagiophthalmosuchus gracilirostris* (NHMUK PV OR 15500); two early-diverging metriorhynchoids *Pelagosaurus typus* (NHMUK PV OR 32599) and *Eoneustes gaudryi* (NHMUK PV R 3263); and three

metriorhynchids, *Thalattosuchus superciliosus* (NHMUK PV R 11999), *Cricosaurus araucanensis* (MLP 72-IV-7-1) and *Cricosaurus schroederi* (MM Pa1). Apart from *Pl. gracilirostris* and *Cri. araucanensis*, which have nearly complete rostra, all the thalattosuchian specimens are missing the anterior portion of the rostrum (comprising the premaxilla and the anterior end of the maxilla) but preserve the portions relevant to this study. All data is available in the Morphsource online repository, under project ID: 000510181 (https://www.morphosource.org/projects/000510181). The files are available at: (https://www.morphosource.org/concern/media/000510291, https://www.morphosource.org/concern/media/000510287, https://www.morphosource.org/concern/media/000510282, https://www.morphosource.org/concern/media/000510274, https://www.morphosource.org/concern/media/000510270, https://www.morphosource.org/concern/media/000510266, https://www.morphosource.org/concern/media/000510261, https://www.morphosource.org/concern/media/000510252, https://www.morphosource.org/concern/media/000510246, https://www.morphosource.org/concern/media/000510242, https://www.morphosource.org/concern/media/000510232, https://www.morphosource.org/concern/media/000510227).

The fossils were segmented manually using Materialise Mimics Innovation Suite (version 24.0, Materialize NV 2021) using the livewire tool. The palatal canals were identified in coronal view as circular or elliptical holes in the palatine bones that bud off the nasal cavity and form a canal oriented anteroventrally which communicates ventrally with the oral cavity. To aid our understanding of palatal vasculature in extant crocodylians, we examined a CT scan of a skull of *Alligator mississippiensis* (OUVC 9757) where the arteries and veins had been injected with a barium-latex contrast medium prior to CT scanning, which created a strong contrast between the vessels and surrounding tissues (see *Porter, Sedlmayr & Witmer, 2016*). The vessels were segmented as one mask by using the threshold segmentation tool on the full scan in Materialize Mimics. The palatal vessels and plexus were then removed from this first mask and segmented using the threshold tool in coronal view at every fifth slice. The 3D interpolate function was then used to fill in the gaps between these slices. The data for this scan is available within Morphosource project ID 000510181, objects 000510189, 000510194, and 000510202 (https://www.morphosource.org/concern/media/000510189, https://www.morphosource.org/concern/media/000510194, and https://www.morphosource.org/concern/media/000510202).

To compare the osteology of the crania, the thalattosuchians were compared to CT scans of 17 extant crocodylians from 11 species (Figs. 2C, 2D and Fig. S1). We included two species of alligatorid, *Alligator mississippiensis* (OUVC 10606, OUVC 9761, OUVC 11415, TMM M-983, and USNM 211233) and *Caiman crocodilus* (FMNH 73711); seven species of crocodylid, *Crocodylus acutus* (FMNH 59071), *Cro. johnstoni* (TMM M-6807), *Cro. moreletii* (TMM M-4980), *Cro. porosus* (OUVC 10899), *Cro. rhombifer* (MNB AB50.071), *Mecistops cataphractus* (TMM M-3529), and *Osteolaemus tetraspis* (FMNH 98936); and two species of gavialid, *Gavialis gangeticus* (TMM M-5490 and UF herp 118998) and *Tomistoma schlegelii* (TMM M-6342 and USNM 211322). Our sample spanned the entire range of crocodylian snout shapes, from broad platyrostral to tubular longirostrine (see Fig. S1). Finally, we included multiple specimens of *A. mississippiensis*,

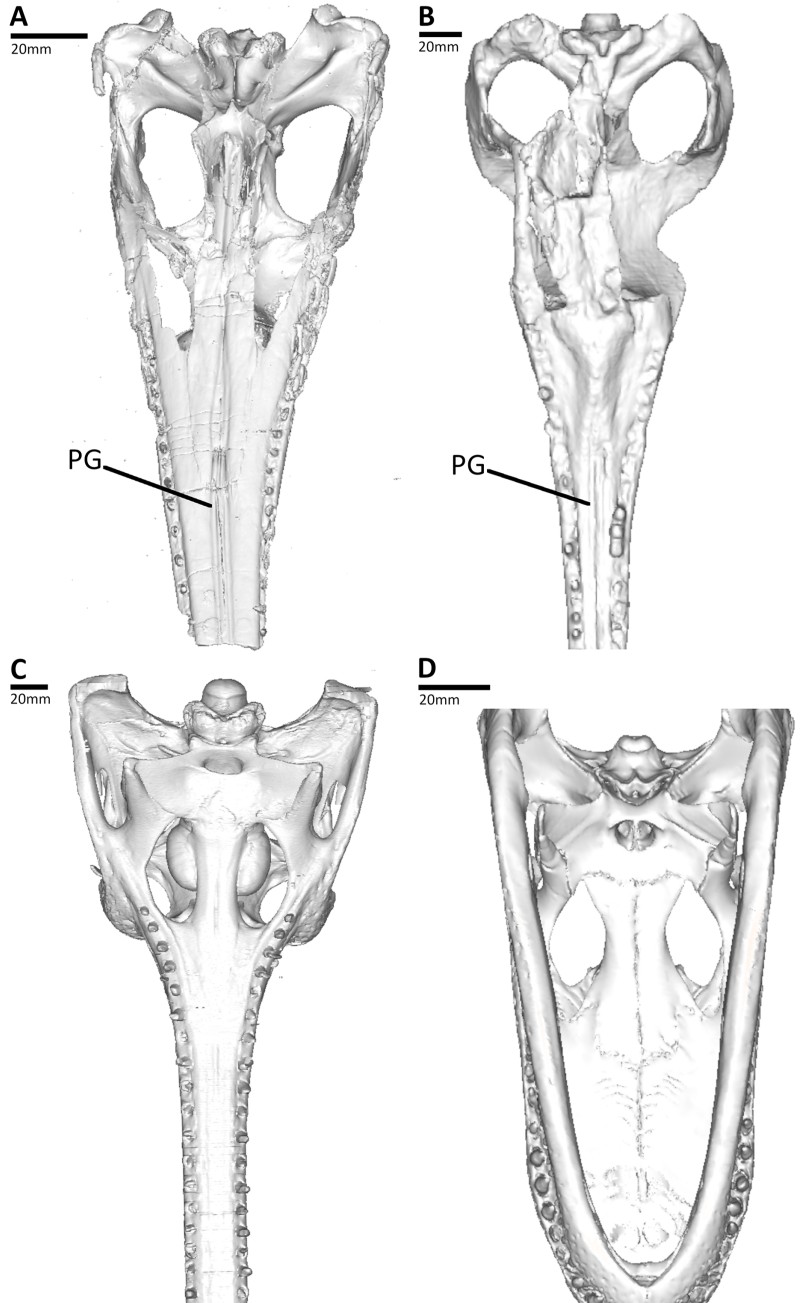

**Figure 2 Comparison between the thalattosuchian and extant crocodylians studied, CT reconstructions of the skulls shown in palatal view.** (A) NHMUK PV OR 32599, the early-diverging metriorhynchoid *Pelagosaurus typus*; (B) MLP 72-IV-7-1, the metriorhynchid *Cricosaurus araucanensis*; (C) UF herp 118998, the gavialid *Gavialis gangeticus*; (D) TMM M983, the alligatorid *Alligator mississippiensis*. Abbreviations: PG, palatal groove.

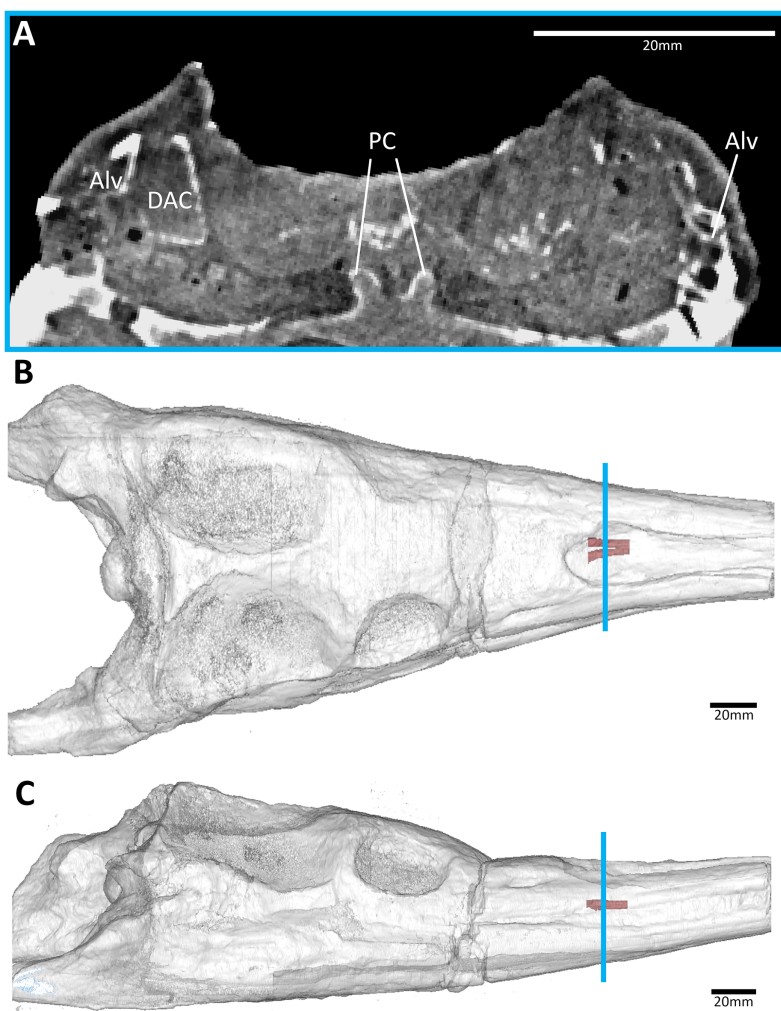

**Figure 3 The early-diverging teleosauroid *Plagiophthalmosuchus gracilirostris* (NHMUK PV OR 15500), from the early Toarcian of the UK.** (A) Snout coronal view showing the position of the palatal canals. Three-dimensional reconstruction of the skull in (B) dorsal, and (C) lateral view, both showing the palatal canals in red and the CT slice of (A) shown in blue. Abbreviations: Alv, alveolus; DAC, dorsal alveolar canal; PC, palatal canal.

*G. gangeticus* and *To. schlegelii* to ascertain whether the presence of palatal grooves was impacted by ontogeny.

## RESULTS

All thalattosuchian skulls in our sample have paired osseous canals that are enclosed by the palatines (Figs. 3–8). These canals are oriented anteroventrally connecting the nasal cavity to the oral cavity (Figs. 3–8). They open into the oral cavity *via* paired foramina at the posterior terminus of the palatal grooves (best seen in *Pelagosaurus typus*, Fig. 1B). In the semi-aquatic thalattosuchians (*i.e.*, the teleosauroid *Plagiophthalmosuchus* and the early-diverging metriorhynchoid *Pelagosaurus*), the canals are almost horizontal when seen in lateral view (Figs. 3C and 4C) and converge at a shallow angle when seen in dorsal view (Figs. 3B and 4B). In contrast, *Eoneustes* and the fully aquatic metriorhynchids have

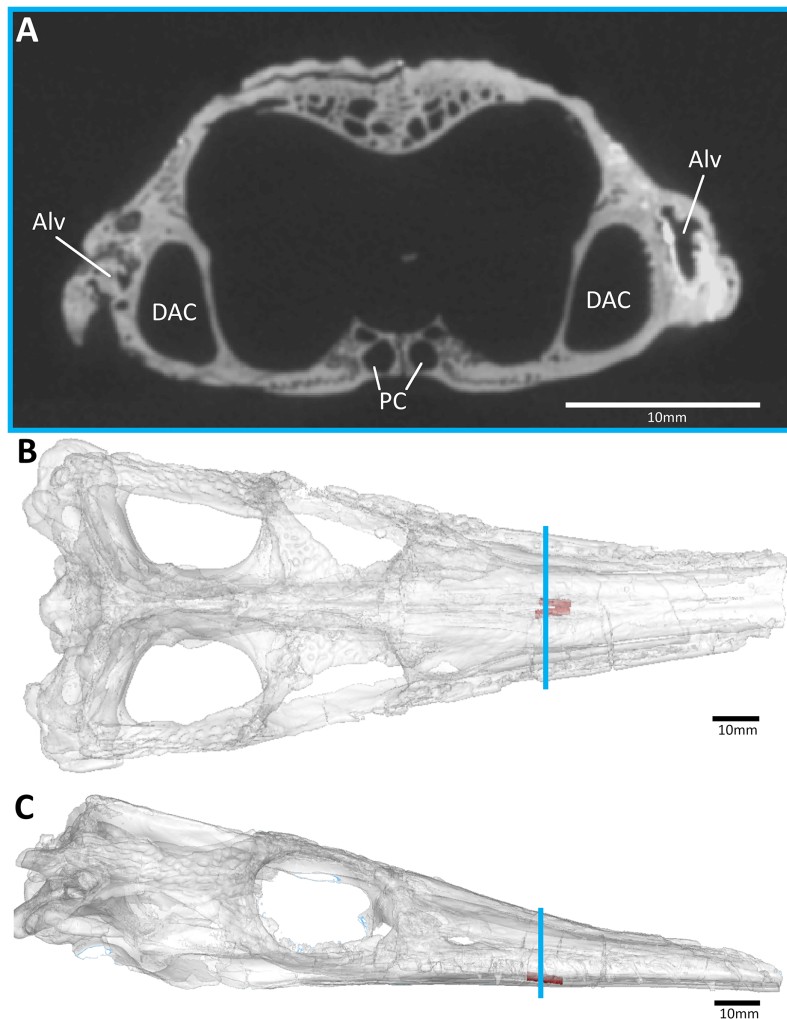

**Figure 4 The early-diverging metriorhynchoid *Pelagosaurus typus* (NHMUK PV OR 32599) referred specimen, early Toarcian of France.** (A) Snout coronal view showing the position of the palatal canals. Three-dimensional reconstruction of the skull in (B) dorsal, and (C) lateral view, both showing the palatal canals in red and the CT slice of (A) shown in blue. Abbreviations: Alv, alveolus; DAC, dorsal alveolar canal; PC, palatal canal.                             

palatal canals that are noticeably angled anteroventrally when seen in lateral view (Figs. 5C, 6C, 7C and 8C). The metriorhynchids also have canals that converge anteriorly at a greater angle (Figs. 6B, 7B and 8B).

The thalattosuchian skulls in our sample also had paired anteroposteriorly aligned parasagittal grooves on the palatal surface of the palatines and maxilla (= palatal grooves; Figs. 1, 2A and 2B). These grooves are a synapomorphy of Thalattosuchia, and are present in all examined metriorhynchoids and in Early Jurassic teleosauroids (*e.g., Andrews, 1913; Parrilla-Bel et al., 2013; Foffa & Young, 2014; Johnson et al., 2019; Johnson, Young & Brusatte, 2020; Aiglstorfer, Havlik & Herrera, 2020; Hua, 2020; Young et al., 2020a, 2021*; Figs. 1, 2A and 2B). Amongst teleosauroids, the grooves appear to have become progressively shallower. While the grooves are clearly present in Early Jurassic

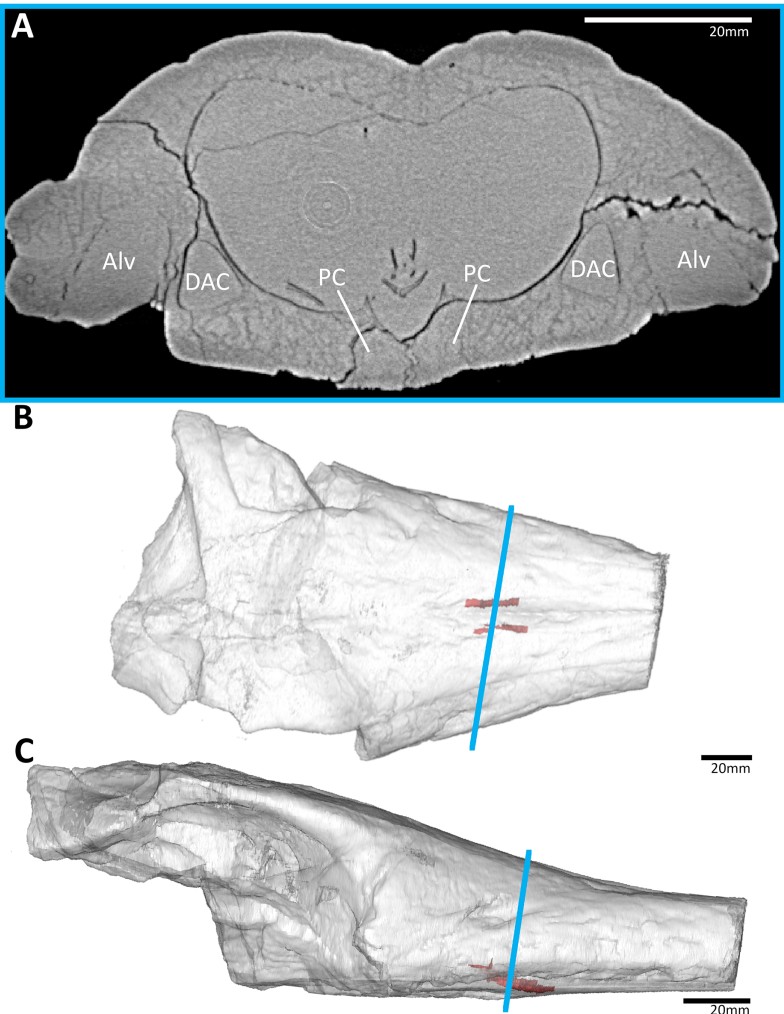

**Figure 5** **The early-diverging metriorhynchoid *Eoneustes gaudryi* (NHMUK PV R 3263) holotype, Bathonian of France.** (A) Snout coronal view showing the position of the palatal canals. Three-dimensional reconstruction of the skull in (B) dorsal, and (C) lateral view, both showing the palatal canals in red and the CT slice of (A) shown in blue. Abbreviations: Alv, alveolus; DAC, dorsal alveolar canal; PC, palatal canal.

teleosauroids (such as the Chinese teleosaurid, see Fig. 1A and *Johnson, Young & Brusatte, 2020*: figure 5; and the teleosaurine teleosaurid *Platysuchus*, see *Johnson et al., 2019*: figure 9), in Middle and Late Jurassic taxa the grooves are harder to discern, being extremely shallow in the aeolodontin teleosaurid *Sericodon* (*Johnson, Young & Brusatte, 2020*), and potentially absent or vestigial in the aeolodontin teleosaurid *Bathysuchus* (see *Foffa et al., 2019*: figure 2) and the machimosaurin machimosaurids *Lemmysuchus* and *Machimosaurus* (see *Johnson et al., 2018*: figure 4; *Johnson, Young & Brusatte, 2020*: figure 27). In *Plagiophthalmosuchus*, the Chinese teleosaurid and *Pelagosaurus*, the grooves are close to the skull midline and remain parallel on the palatines and for most of the maxilla (diverging only in the anterior-most region of the maxilla) (see *Andrews, 1913*; *Pierce & Benton, 2006*; *Johnson et al., 2019*; *Johnson, Young & Brusatte, 2020*; Figs. 1A, 1B,

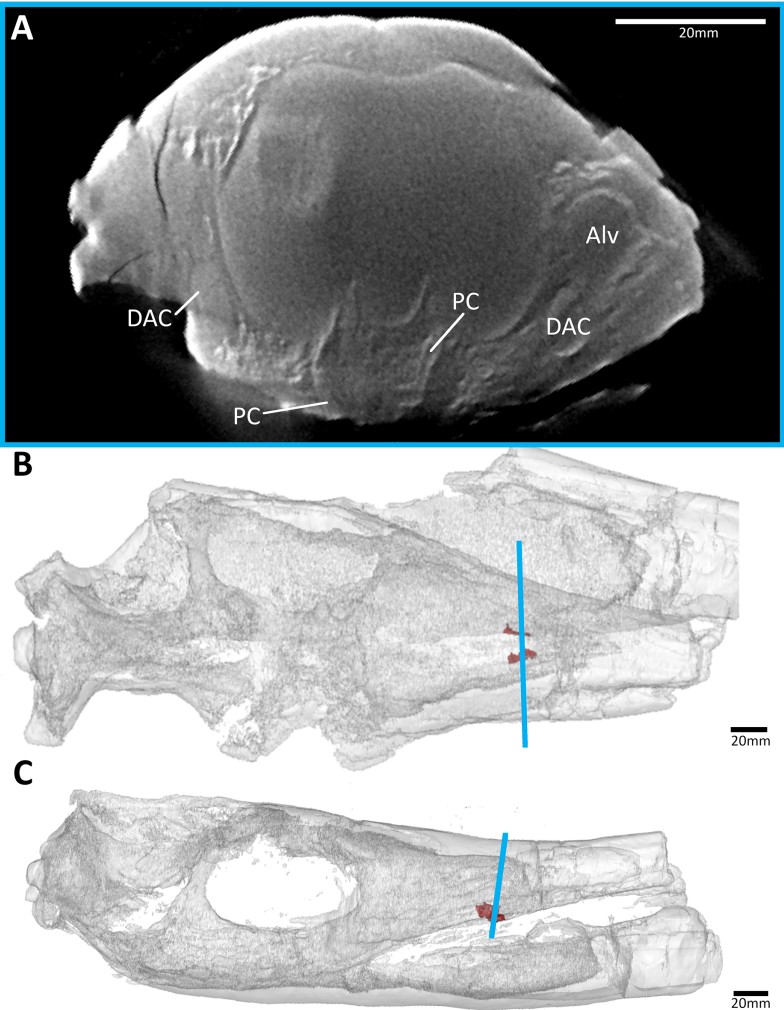

**Figure 6 The metriorhynchid *Thalattosuchus superciliosus* (NHMUK PV R 11999) referred specimen, middle Callovian of the UK.** (A) Snout coronal view showing the position of the palatal canals. Three-dimensional reconstruction of the skull in (B) dorsal, and (C) lateral view, both showing the palatal canals in red and the CT slice of (A) shown in blue. Abbreviations: Alv, alveolus; DAC, dorsal alveolar canal; PC, palatal canal.

and 2A). In metriorhynchids however, the grooves diverge at the anterior palatines, and on the maxilla the grooves become largely parallel but are much more widely separated than in non-metriorhynchid thalattosuchians (see *Andrews, 1913*; *Parrilla-Bel et al., 2013*; *Foffa & Young, 2014*; *Young et al., 2020a*, *2021*; Fig. 1). The shift in morphology occurs gradually within Metriorhynchoidea, as the palatal grooves become more widely spaced relative to the maxillary midline in the early-diverging metriorhynchoids *Teleidosaurus*, *Opisuchus* and *Eoneustes*, being intermediate between *Pelagosaurus* and metriorhynchids (see *Aiglstorfer, Havlik & Herrera, 2020*; *Hua, 2020*; NHMUK PV R 3263).

In contrast, all of the extant crocodylian skulls in our sample lacked the external palatal grooves (Figs. 2C, 2D, and Fig. S1) and the internal canals (Figs. S2 and S3). This was true for alligatorids (*Alligator mississippiensis* and *Caiman crocodilus*), crocodylids (*Crocodylus*

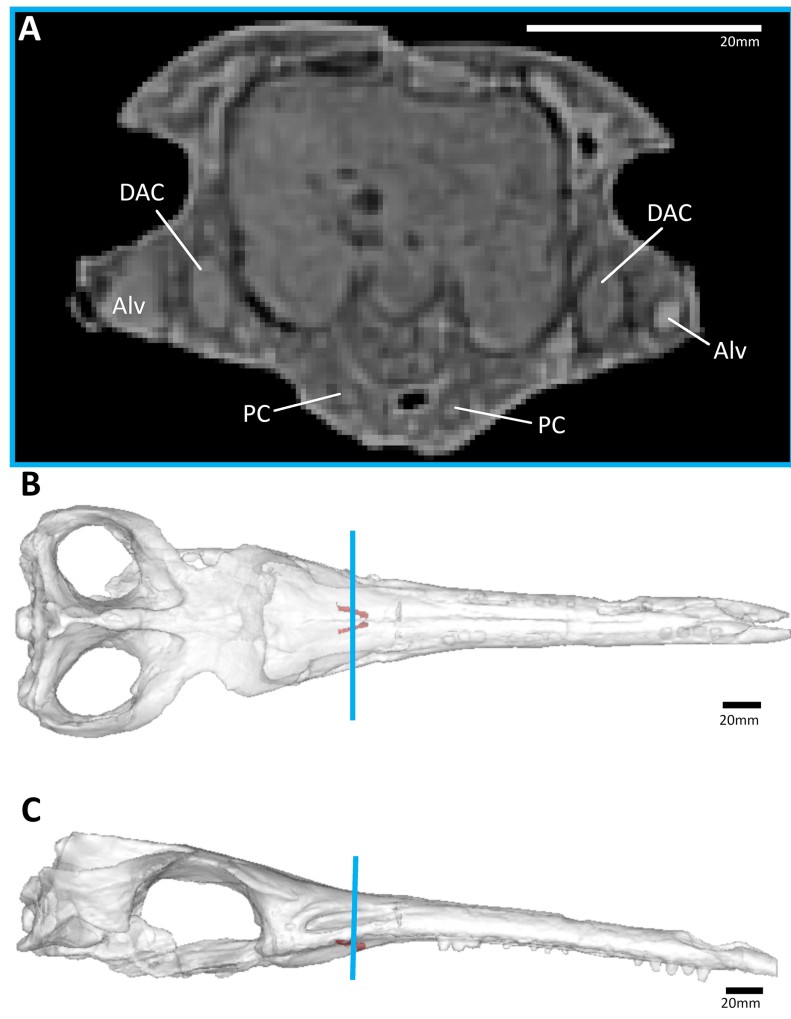

**Figure 7 The metriorhynchid *Cricosaurus araucanensis* (MLP 72-IV-7-1) holotype, Tithonian of Argentina.** (A) Snout coronal view showing the position of the palatal canals. Three-dimensional reconstruction of the skull in (B) dorsal, and (C) lateral view, both showing the palatal canals in red and the CT slice of (A) shown in blue. Abbreviations: Alv, alveolus; DAC, dorsal alveolar canal; PC, palatal canal.

*acutus, Cro. johnstoni, Cro. moreletii, Cro. porosus, Cro. rhombifer, Mecistops cataphractus,* and *Osteolaemus tetraspis*) and gavialids (*Gavialis gangeticus* and *Tomistoma schlegelii*). Moreover, the grooves and canals are not present in any of the different ontogenetic stages we examined, including the hatchling (Fig. S1A), juveniles (Figs. S1B–S1D and S1J), subadults (Figs. S1F, S1K, S1L, S1N and S1P) and adults (Figs. S1E, S1M, S1O, and S1Q). Based on our sample of thalattosuchians and extant crocodylians we posit that the osseous palatal canals and the external grooves are linked. Both structures are continuous, and are only found to co-occur (*i.e.*, skulls lacking palatal grooves also lack internal palatal canals, and skulls which have palatal grooves also have internal palatal canals). We cannot find evidence of palatal grooves in any extant or extinct non-thalattosuchian crocodylomorph based on first-hand examination of specimens (Table S2) or from the literature (Table S3).

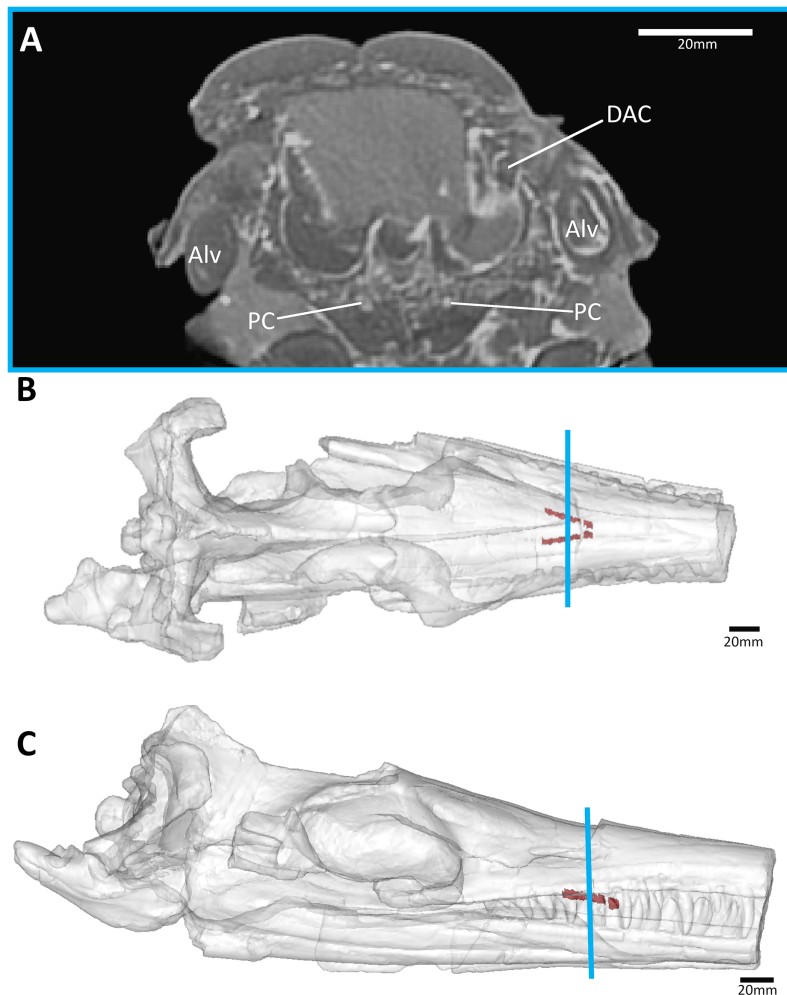

**Figure 8 The metriorhynchid *Cricosaurus schroederi* (MM Pa1), from the early Valanginian of Germany.** (A) Snout coronal view showing the position of the palatal canals. Three-dimensional reconstruction of the skull in (B) dorsal, and (C) lateral view, both showing the palatal canals in red and the CT slice of (A) shown in blue. Abbreviations: Alv, alveolus; DAC, dorsal alveolar canal; PC, palatal canal.

## DISCUSSION

### Palatal structures in crocodylomorpha

The presence of palatal grooves is one of the defining characteristics of Thalattosuchia (*Andrews, 1913*; *Parrilla-Bel et al., 2013*; *Foffa & Young, 2014*; *Johnson et al., 2019*; *Johnson, Young & Brusatte, 2020*; *Aiglstorfer, Havlik & Herrera, 2020*; *Young et al., 2020a, 2021*). As we note herein, these grooves are not found in extant crocodylians, irrespective of their ontogenetic stage. Given that no other mesoeucrocodylian taxon with a maxillopalatine secondary palate has been observed to have palatal grooves, we posit that they are synapomorphies of Thalattosuchia. This is in agreement with phylogenetic analyses that have found these features to be explicit thalattosuchian synapomorphies (*e.g.*, *Johnson, Young & Brusatte, 2020*; *Young et al., 2020a, 2021*).

Some notosuchians are known to have prominent palatal openings. Paired foramina are present at the maxilla-palatine boundary in species referred to Notosuchidae and Sphagesauridae, namely: *Notosuchus terrestris* (*Andrade & Bertini, 2008a*; *Barrios et al., 2018*), *Mariliasuchus amarali* (*Andrade, Bertini & Pinheiro, 2006*; *Zaher et al., 2006*), *Llanosuchus tamaensis* (*Fiorelli et al., 2016*), and *Caipirasuchus mineirus* and *Caipirasuchus stenognathus* (*Pol et al., 2014*; *Martinelli et al., 2018*). Curiously, these openings are absent in some sphagesaurids, including *Caipirasuchus montealtensis* and *Caipirasuchus paulistanus* (*Andrade & Bertini, 2008b*; *Iori et al., 2013*; *Pol et al., 2014*; *Martinelli et al., 2018*), and in *Sphagesaurus huenei* and *Yacarerani boliviensis* (*Pol et al., 2014*). Referred to as 'maxillo-palatine fenestrae' in the notosuchian literature, openings are often larger than the foramina described herein for thalattosuchians, especially for *Caipirasuchus mineirus* where these openings are very large (see *Martinelli et al., 2018*: figures 17 and 32). The 'maxillo-palatine fenestrae' seen in notosuchids and some sphagesaurids are in a similar position to the palatal foramina of thalattosuchians (see Fig. 1B). However, the notosuchian fenestrae are noticeably larger, and lack associated palatal grooves. Moreover, it appears that the 'maxillo-palatine fenestrae' seen in notosuchids and in two species of the sphagesaurid genus *Caipirasuchus* evolved independently (given their absence in all other sphagesaurid species). As such, the palatal foramina of thalattosuchians and the 'maxillo-palatine fenestrae' of sphagesaurian notosuchians do not share a common origin.

In the notosuchian *Simosuchus clarki*, *Kley et al. (2010*: 38, figures 3B and 8F) described paired palatal fossae on the anterior palatal rami of the maxilla, at the premaxilla-maxilla boundary. Within each deep fossa is a palatal foramen. However, given their anterior position and lack of palatal grooves we do not consider them to be homologous to the palatal canals found in thalattosuchians. In the same location, large foramina are also found in the allodaposuchid *Lohuecosuchus megadontos*, however there is no surrounding fossa (*Narváez et al., 2015*). Interestingly, mid-way along the maxilla there are paired foramina close to the skull midline in *Lohuecosuchus megadontos* (*Narváez et al., 2015*). However, these palatal foramina are not found in any other species of allodaposuchid (*Narváez et al., 2015*: 25).

Shartegosuchoids are another clade of fossil crocodylomorphs with a curious palate. At the skull midline, there is often a large opening referred to as the anterior palatal fenestra (*Dollman et al., 2018*; *Dollman & Choiniere, 2022*). In early-diverging shartegosuchoids, the anterior palatal fenestra, the nasopharyngeal duct and the internal choana are all continuous; however, in *Fruitachampsa* and *Shartegosuchus* the anterior palatal fenestra became isolated from the internal choana due to the formation of secondary bony palatal shelves (*Dollman & Choiniere, 2022*: 9). The morphology of the shartegosuchoid palatal fenestra is radically different from the palatal foramina seen in thalattosuchians, and the shartegosuchoids lack palatal grooves.

## Palatal grooves in aquatic mammals and oral vascularisation

While no crocodylomorph clade shares the paired longitudinal palatal grooves seen in Thalattosuchia, curiously fossil and extant cetaceans do. A very similar morphology is

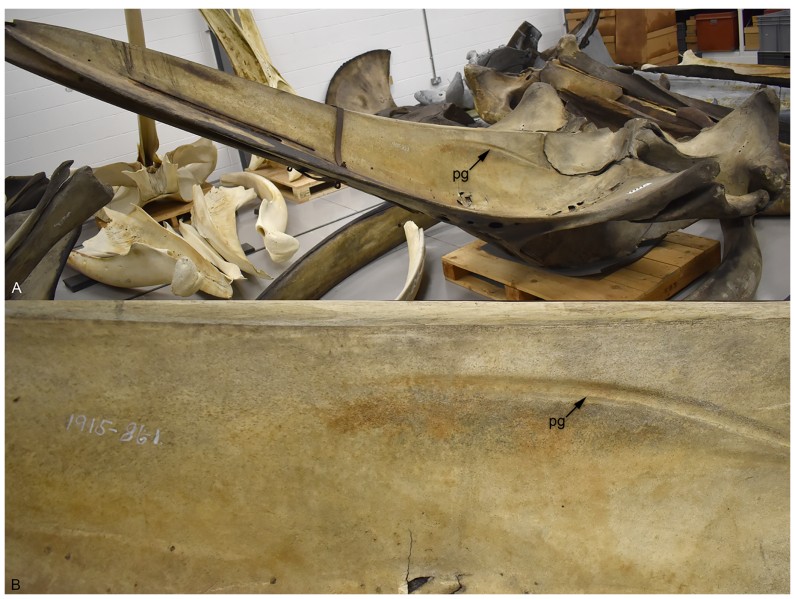

**Figure 9 The extant humpback whale (*Megaptera novaeangliae*) NMS.Z.1915.86.2—note the 86.1 written on the specimen is an error.** (A) Skull showing the palate, due to size the skull it is shown at an angle; (B) a close-up on the right palatal groove. Abbreviations: PG, palatal groove.

present in the semi-aquatic remingtonocetid *Remingtonocetus harudiensis* (*Bajpai, Thewissen & Conley, 2011*: figure 1.3), the semi-aquatic protocetid *Aegyptocetus tarfa* (*Bianucci & Gingerich, 2011*: figure 3), and in fully aquatic forms, including the early-diverging mysticete *Aetiocetus weltoni* (*Ekdale & Deméré, 2022*: figure 2A), the early-diverging odontocetes *Simocetus rayi* (*Fordyce, 2002*: figure 4) and *Echovenator sanderi* (*Churchill et al., 2016*: figures 1F and 1G), and the beluga-like odontocete *Bohaskaia monodontoides* (*Vélez-Juarbe & Pyenson, 2012*: figure 3). The same morphology has also been described and figured for the extant gray whale (*Eschrichtus robustus*) and finback whale (*Balaenoptera physalus*) (see *Ekdale, Deméré & Berta, 2015*), and is also present in the humpback whale (*Megaptera novaeangliae*) (Fig. 9). There are four striking parallels between thalattosuchians and cetaceans: (1) the presence of anteroposteriorly aligned (longitudinal) grooves, present along most of the maxilla with their posterior terminus either on the palatines (as in thalattosuchians) or at the maxilla-palatine suture (cetaceans); (2) the longitudinal grooves have a large foramen at their posterior terminus; (3) in both clades the morphology is present in both semi-aquatic and fully aquatic forms; and (4) the grooves are closer to the skull midline in the semi-aquatic forms (see *Pelagosaurus* herein and *Remingtonocetus* in *Bajpai, Thewissen & Conley, 2011*), whereas in the fully aquatic forms the grooves are much more widely spaced (see the metriorhynchids herein, and *Aetiocetus* in *Ekdale & Deméré, 2022*; *Simocetus* in *Fordyce, 2002*; *Echovenator* in *Churchill et al., 2016* and *Bohaskaia* in *Vélez-Juarbe & Pyenson, 2012*). Intermediate morphologies also appear in cetacean evolution, such as in *Aegyptocetus* (*Bianucci & Gingerich, 2011*).

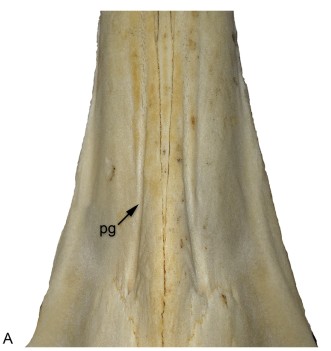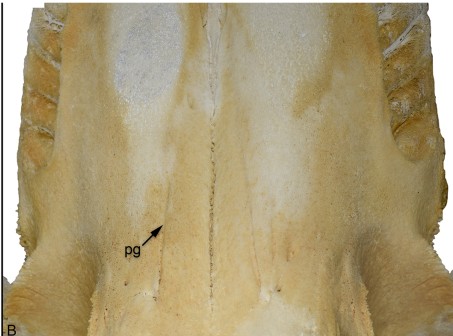

**Figure 10 Comparison of the palatal grooves in different extant odontocete cetaceans, skulls shown in palatal view.** (A) Cuvier's beaked whale (*Ziphius cavirostris*) NMS 2020.9.26; (B) killer whale (*Orcinus orca*) NMS Z.2015.179. Abbreviations: PG, palatal groove.

In extant whales, the greater (or descending) palatine artery exits through the palatal foramen and continues anteriorly *via* the longitudinal groove/sulcus (*Deméré et al., 2008*; *Ekdale, Deméré & Berta, 2015*). This has also been hypothesised for fossil cetaceans (*e.g.*, *Bajpai, Thewissen & Conley, 2011*; *Vélez-Juarbe & Pyenson, 2012*; *Ekdale & Deméré, 2022*). Although there has been a long discussion on whether the greater palatine artery is associated with the evolution of baleen in mysticetes, this hypothesis seems to have been falsified (see *Ekdale, Deméré & Berta, 2015*; *Ekdale & Deméré, 2022*). Two further hypotheses have been suggested for the expansion of the palatine vasculature in cetaceans, positing that it is either a consequence of rostral elongation (*Ichishima et al., 2008*) or for thermoregulation (*Ekdale, Deméré & Berta, 2015*). *Ekdale, Deméré & Berta (2015*: 699), however, noted that similar structures are not found in other mammals with elongate snouts (although the palatine foramina, and some form of palatal grooves, are). Within Crocodylomorpha there are numerous long-snouted groups, both extinct and extant, but none show evidence of palatal grooves. Moreover, among extant species long-snouted taxa do not have expanded rostral vasculature compared to broader snouted species (*e.g.*, *Bowman et al., 2022*).

Mysticetes have highly vascularised oral cavities, with the mouth being an important site for thermoregulation (*e.g.*, *Ford & Kraus, 1992*; *Werth, 2007*; *Ford, Werth & George, 2013*; *Ekdale, Deméré & Berta, 2015*). This is unsurprising given that mysticetes bulk filter feed, which involves the mouth being repeatedly exposed to (often cold) sea water. However, odontocetes seem to lack vascular adaptations for thermoregulation within the oral cavity (*Werth, 2007*). This is supported by the palatine foramen being greatly reduced, or almost closed, in extant delphinoid odontocetes (although the foramina are greatly enlarged in the fossil genus *Odobenocetops*, see *de Muizon, Domning & Ketten, 2002*), although the grooves are present in the killer whale (*Orcinus orca*) and Cuvier's beaked whale (*Ziphius cavirostris*) (Fig. 10). *Werth (2007)* suggested that for odontocetes there was either less need to prevent oral heat loss, or that other regions of the body were more important sites for thermoregulation.

During their land-to-sea transition, pinnipedimorphs (seals and their close fossil relatives) evolved a similar morphology (Fig. S4). In early-diverging forms such as *Enalioarctos*, the palatal grooves originating from the palatine foramina are relatively short (*Berta, 1991*). During phocid ('true seals') evolution, however, the grooves became increasingly broader and more elongated (*Dewaele, Lambert & Louwye, 2018*; *Rule et al., 2020*; *Koretsky & Rahmat, 2021*).

Many other amniote groups have a venous plexus within the soft tissues of the palate. In birds, the palatal plexus and the rete ophthalmicum help maintain eye and brain temperature (*Kilgore, Boggs & Birchard, 1979*; *Midtgård, 1983*, *1984*; *Porter & Witmer, 2016*), while in extant crocodylians there is an extensive palatal plexus (*Porter, Sedlmayr & Witmer, 2016*). In extant archosaurs the palatal plexus is supplied by the palatine artery (Figs. S5 and S6); however, the palatine arteries travel through the soft tissue of the secondary palate (see *Porter & Witmer, 2016*; *Porter, Sedlmayr & Witmer, 2016*), unlike in cetaceans where they pass through the bony palate. Moreover, in extant archosaurs the palatine arteries are situated laterally in the rostrum (see Fig. S5; *Porter & Witmer, 2016*; *Porter, Sedlmayr & Witmer, 2016*), not medially as in cetaceans. We propose an osteological correlate for the palatine vessels in thalattosuchians: the groove that originates at the anterior margin of the suborbital fenestra (Fig. 1: SOG). This groove is consistent with location of the palatine vessels in extant crocodylians (*Porter, Sedlmayr & Witmer, 2016*).

Based on the striking similarity between thalattosuchian palatal canal/groove system and those of cetaceans (particularly the fossil semi-aquatic and aquatic species), and the known routes and positions of extant crocodylian cranial vasculature, we hypothesise the following:

1. The thalattosuchian palatal canal/groove system transmitted the medial nasal vessels (artery and vein) or a novel branch thereof, and possibly also some of the rostral nerves. In all extant diapsids, the medial nasal vessels branch off from nasal vessels at the posterodorsal aspect of the nasal cavity. The medial nasal vessels then descend anteroventrally on either side of the median cartilaginous internasal septum to run on the floor of the nasal cavity (*e.g.*, Figs. S5 and S6; *Porter & Witmer, 2015*, *2016*; *Porter, Sedlmayr & Witmer, 2016*). Therefore, the paramedian/parasagittal position of the palatal canal/groove system in thalattosuchians is consistent with the medial nasal vessels.

2. Early in thalattosuchian evolution, the medial nasal vessels (or a ventral branch thereof) pierced the bony palate to emerge on to the roof of the oral cavity. If thalattosuchians are a non-crocodyliform clade (*e.g.*, see *Wilberg et al., 2023*) then the formation of the secondary bony palate occurred independently within Thalattosuchia, and therefore nasal vessels would have been enclosed by the palatines during the formation of the maxillopalatine palate.

3. The medial nasal vessels that entered the oral cavity anastomosed with the palatal vascular plexus (which are supplied by the palatine vessels).

4. The large internal osseous canals represent a hypertrophy of the medial nasal vessels.

5. A novel heat exchange pathway was created by linking the palatal plexus to medial nasal vessels. In extant crocodylians, the medial nasal vessels communicate with the encephalic arteries and veins *via* the ethmoid vessels (*Porter, Sedlmayr & Witmer, 2016*). The palatal vascular plexus is a critical location of thermal exchange in extant crocodylians (*Porter, Sedlmayr & Witmer, 2016*). While the palatal plexus is not thought to have a substantial role in thermoregulation of the brain in extant crocodylians, based on our proposed vascular pathway, the palatal plexus would have moderated brain temperatures of thalattosuchians *via* the ethmoid vessels.

## Increased cephalic blood volume in Thalattosuchia

A novel heat exchange pathway to help maintain brain and eye temperatures would have been greatly beneficial for Metriorhynchidae. Not only did metriorhynchids have an elevated metabolism (possibly a poorly homeothermic form of endothermy, see *Séon et al., 2020*), but they had expanded cerebral hemispheres and orbits relative to extant crocodylians and other thalattosuchians (*e.g.*, see *Young et al., 2010*; *Herrera, Leardi & Fernández, 2018*; *Schwab et al., 2021*). An obvious question is why would semi-aquatic thalattosuchians also have had a novel heat exchange pathway? One of the defining features of Thalattosuchia is the enlarged cerebral carotid foramina on the occipital surface of the cranium, being found in both semi-aquatic and fully aquatic species (*Andrews, 1913*; *Pierce & Benton, 2006*; *Jouve, 2009*; *Pol & Gasparini, 2009*; *Fernández et al., 2011*; *Young et al., 2012*, *2013*, *2020b*; *Herrera & Vennari, 2015*; *Brusatte et al., 2016*; *Johnson, Young & Brusatte, 2020*). Note, the cerebral carotid foramina become even larger in the clade *Zoneait* + Metriorhynchidae (*Wilberg, 2015*; *Herrera, Leardi & Fernández, 2018*), while they become smaller in some freshwater teleosauroids (*Herrera, Leardi & Fernández, 2018*). In mammals, larger encephalic arteries are associated with higher rates of blood flow, as flow (perfusion) is proportional to the radius of the arterial lumen raised to an exponent of approximately 2.5 (*Seymour et al., 2019*). In extant crocodylians, the cerebral carotid arteries supply blood to the brain, eyes, nasal cavities, and the rostral sinuses (*Porter, Sedlmayr & Witmer, 2016*). Therefore, it is possible that these vessels supplied a greater volume of blood to these regions in thalattosuchians compared to extant crocodylians.

Further, these enlarged foramina do not represent the full extent of vascular hypertrophy observed in thalattosuchian skulls. The cerebral carotid vessels enter the greatly enlarged pituitary fossa chamber, another thalattosuchian synapomorphy, which in extant crocodylians houses the cavernous venous sinus (*Porter, Sedlmayr & Witmer, 2016*) and was possibly hypertrophied in thalattosuchians. Exiting the anterior margin of the pituitary fossa chamber are two ossified canals thought to transmit the orbital arteries (*Brusatte et al., 2016*), with these canals being almost as wide as the cerebral carotid canals (*Brusatte et al., 2016*; *Pierce, Williams & Benson, 2017*; *Herrera, Leardi & Fernández, 2018*; *Wilberg et al., 2022*). Within Crocodylomorpha, only thalattosuchians and the dyrosaurid *Rhabdognathus* (*Erb & Turner, 2021*) are known to have the orbital arteries contained within ossified canals. Further, the midbrain and hindbrain of thalattosuchians are very

poorly delineated in their endocasts due to the hypertrophy of the longitudinal and transverse dural venous sinuses, the latter being continuous with the hypertrophied stapedial canals (*Wharton, 2000*; *Fernández et al., 2011*; *Brusatte et al., 2016*; *Pierce, Williams & Benson, 2017*; *Herrera, Leardi & Fernández, 2018*; *Schwab et al., 2021*; *Wilberg et al., 2022*). Collectively, this implies that thalattosuchians had increased encephalic blood volumes and potentially increased perfusion rates relative to extant crocodylians. As such, maintaining stable brain and eye temperatures may have required more extensive heat exchange mechanisms.

Unfortunately, we do not know the timing of these internal changes. All thalattosuchians that have been subject to CT scanning show the same suite of vascular characters outlined above, and the palatal groove/canal system described herein. It is unclear whether encephalic vascular evolution in Thalattosuchia was stepwise and gradual, or whether one of these characteristics was a 'key adaptation' that triggered rapid change within the thalattosuchian skull. Only new fossil discoveries, of taxa basal to the teleosauroid-metriorhynchoid split, will allow us to understand this radical reorganisation.

Regardless of what selection pressures drove basal thalattosuchians to evolve these encephalic vascular characteristics, we posit that within Metriorhynchoidea, as the clade became increasingly aquatic, these characteristics made possible the evolution of larger orbits, larger cerebral hemispheres, and an elevated metabolism. An elevated metabolism and a pathway to help maintain stable brain and eye temperatures would also have made feeding below the thermocline viable, especially in a group considered to be primarily vision-based hunters (*Massare, 1988*; *Martill et al., 1994*; *Young et al., 2010*; *Bowman et al., 2022*). Isotopic analyses suggest that belemnites lived below the thermocline during the Jurassic (*e.g.*, *Jenkyns et al., 2012*; *Xu et al., 2018*), and an abundance of belemnite hooklets have been found within the body cavity of Middle Jurassic metriorhynchids from the Oxford Clay Formation of the UK (*Martill, 1986*). While the evolution of hypertrophied salt glands has been cited as an example of how physiological changes expanded the metriorhynchid prey envelope, to include osmoconforming species (*Fernández & Gasparini, 2000*, *2008*; *Cowgill et al., 2022a*), thermophysiological changes were undoubtedly also exceptionally important. The suite of vascular characters outlined herein are unique to thalattosuchians, and no other crocodylomorph clade contained a lineage that evolved to become fully aquatic. Perhaps these changes in cranial vasculature were a necessary precursor for the development of the fully aquatic metriorhynchids.

Well-developed palatal grooves are only found in Early Jurassic teleosauroids (*e.g.*, *Johnson et al., 2019*; *Johnson, Young & Brusatte, 2020*), whereas in Middle and Late Jurassic taxa the grooves became extremely shallow or were possibly absent (*e.g.*, *Johnson et al., 2018*; *Johnson, Young & Brusatte, 2020*; *Foffa et al., 2019*). It is possible that as teleosauroids adapted to a 'sit and wait' ambush ecology, the selection pressures that necessitated maintaining stable brain and eye temperatures were no longer as strong. However, CT scans of Middle and Late Jurassic teleosauroids will be needed to determine whether the internal palatal canals were present or not. It is possible that the grooves themselves simply became shallower in these taxa, perhaps somewhat like in extant

odontocete whales—with the oral cavity becoming a less important site of thermoregulation.

## CONCLUSIONS

Herein we show that the palatal grooves of thalattosuchians were unique within Crocodylomorpha. We cannot find any other crocodylomorph clade that had anteroposteriorly aligned grooves along their maxilla and palatines, and cannot find any evidence that the absence of the grooves is influenced by ontogeny. Based on CT scans of thalattosuchian skulls, these grooves are continuous with a pair of canals which travel through the palatines connecting the oral and nasal cavities. The canals open into the posterior terminus of the grooves *via* foramina (best seen in Fig. 1B). These internal canals are also not present in the CT scans of extant crocodylian skulls.

However, the palatal canals, foramina and grooves are strikingly similar to those of another group, cetaceans. Present in both fossil semi-aquatic species, and fossil and extant fully aquatic species, these structures transmit the greater palatine artery which supplies a palatal venous thermoregulatory structure. Given the convergence in palatal grooves between these groups, we hypothesise that the canals and grooves of thalattosuchians transmitted hypertrophied vasculature. Based on the position of the canal/groove system, the most likely candidate are the medial nasal vessels. Connecting the medial nasal vessels to the palatal vascular plexus would have created a novel heat exchange pathway, one that linked the plexus (an important thermoregulatory site) to the vessels that supply blood to the brain and eyes. As thalattosuchians likely had increased cephalic blood volume and flow rates relative to other crocodylomorphs, a corresponding increase in cephalic thermoregulatory capabilities would be necessary. However, at present we cannot ascertain which came first: increased blood flow (*e.g.*, wider cerebral carotid canal and external foramina), increased blood volume (*e.g.*, orbital canals almost as wide as the carotid canals, and hypertrophied pituitary fossa chamber, transverse dural venous sinuses and stapedial canals), or the medial nasal vessel mediated thermoregulatory pathway. We also do not know the rate and order at which these changes occurred. New fossil discoveries are needed to elucidate thalattosuchian cephalic vascular evolution. Given the peculiar palatal morphologies of shartegosuchoids and some notosuchians, it is possible that novel thermoregulatory pathways also evolved in terrestrial crocodylomorphs.

## ACKNOWLEDGEMENTS

We thank S. Maidment (NHMUK), Z. Timmons (NMS) R. Allain (MNHN), and M. Gasparik and Z. Szentesi (MTM) for providing generous access to the specimens in their care, M. Johnson (Stuttgart) for providing photograph of the Chinese teleosaurid and S. Sachs (Bielefeld) for generously helping with the NMS figures. We thank Eric Wilberg and an anonymous reviewer for their constructive comments.

## INSTITUTIONAL ABBREVIATIONS

**FMNH**     Field Museum of Natural History, Chicago, Illinois, USA
**IVPP**      Institute of Paleontology and Paleoanthropology, Beijing, China

| MLP | Museo de La Plata, La Plata, Argentina |
| MM | Minden Museum, Minden, Germany |
| MNB | National Museum of the Bahamas, Nassau, Bahamas |
| MNHN | Muséum national d'Histoire naturelle, Paris, France |
| MTM | Magyar Természettudományi Múzeum, Budapest, Hungary |
| NMS | National Museum of Scotland, Edinburgh, Scotland, UK |
| NHMUK | Natural History Museum, London, UK |
| OUVC | Ohio University, Vertebrate Collection, Athens, Ohio, USA |
| TMM | Texas Memorial Museum, University of Texas at Austin, Austin, Texas, USA |
| UF | University of Florida, Florida Museum of Natural History, Gainesville, Florida, USA |
| USNM | National Museum of Natural History, Washington DC, USA |

### Funding

This work was supported by a Leverhulme Trust Research Project (Grant number RPG-2017-167). Mark T Young received support for his collection visits to Paris (FR-TAF-4021) and Budapest (HU-TAF-6505) from the SYNTHESYS project (http://www.synthesys.info/), which is financed by the European Community Research Infrastructure Action under the FP7 'Capacities' programme. Lawrence M. Witmer is supported by the United States National Science Foundation (IOB-0517257, IOS-1050154, IOS-1456503) and the Swedish Research Council (2021-02973). Yanina Herrera is supported by the ANPCyT (PICT 2020–2067) and CONICET (PIP 2844). The funders had no role in study design, data collection and analysis, decision to publish, or preparation of the manuscript.

### Grant Disclosures

The following grant information was disclosed by the authors:
Leverhulme Trust Research Project: RPG-2017-167.
European Community Research Infrastructure Action: Paris (FR-TAF-4021) and Budapest (HU-TAF-6505).
United States National Science Foundation: IOB-0517257, IOS-1050154, IOS-1456503.
Swedish Research Council: 2021-02973.
ANPCyT: PICT 2020-2067.
CONICET: PIP 2844.

### Competing Interests

Mark T. Young is an Academic Editor for PeerJ.

### Author Contributions

- Mark T. Young conceived and designed the experiments, performed the experiments, analyzed the data, prepared figures and/or tables, authored or reviewed drafts of the article, and approved the final draft.

- Charlotte I. W. Bowman conceived and designed the experiments, performed the experiments, analyzed the data, prepared figures and/or tables, authored or reviewed drafts of the article, and approved the final draft.
- Arthur Erb performed the experiments, analyzed the data, prepared figures and/or tables, authored or reviewed drafts of the article, and approved the final draft.
- Julia A. Schwab analyzed the data, authored or reviewed drafts of the article, and approved the final draft.
- Lawrence M. Witmer analyzed the data, authored or reviewed drafts of the article, and approved the final draft.
- Yanina Herrera analyzed the data, authored or reviewed drafts of the article, and approved the final draft.
- Stephen L. Brusatte analyzed the data, authored or reviewed drafts of the article, and approved the final draft.

## Data Availability

All data is available in the Morphsource online repository, under project ID: 000510181 (https://www.morphosource.org/projects/000510181).

- Media 000510291: Segmentation Of The Palatal Canals (Internal Rostral Canals) [Mesh] [CT], DOI 10.17602/M2/M510291
- Media 000510287: Skull [Mesh] [CT] DOI 10.17602/M2/M510287
- Media 000510282: Segmentation Of The Palatal Canals (Internal Rostral Canals) [Mesh] [CT] DOI 10.17602/M2/M510282
- Media 000510274: Skull [Mesh] [CT] DOI 10.17602/M2/M510274
- Media 000510270: Segmentation Of The Palatal Canals (Internal Rostral Canals) [Mesh] [CT] DOI 10.17602/M2/M510270
- Media 000510266: Skull [Mesh] [CT] DOI 10.17602/M2/M510266
- Media 000510261: Segmentation Of The Palatal Canals (Internal Rostral Canals) [Mesh] [CT] DOI 10.17602/M2/M510261
- Media 000510252: Skull [Mesh] [CT] DOI 10.17602/M2/M510252
- Media 000510246: Segmentation Of The Palatal Canals (Internal Rostral Canals) [Mesh] [CT] DOI 10.17602/M2/M510246
- Media 000510242: Skull [Mesh] [CT] DOI 10.17602/M2/M510242
- Media 000510232: Internal Palatal Canals (Rostrum) [Mesh] [CT] DOI 10.17602/M2/M510232
- Media 000510227: Partial Cranial Rostrum [Mesh] [CT] DOI 10.17602/M2/M510227
- Media 000510202: Segementation Of The Internal Venous System [Mesh] [CT] DOI 10.17602/M2/M510202
- Media 000510194: Segementation Of The Palatal Venous Plexus [Mesh] [CT] DOI 10.17602/M2/M510194
- Media 000510189: Segementation Of The Nasal Venous Vessels [Mesh] [CT] DOI 10.17602/M2/M510189

## Supplemental Information

Supplemental information for this article can be found online at http://dx.doi.org/10.7717/peerj.15353#supplemental-information.

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
