# Peer review of "Evidence for a novel cranial thermoregulatory pathway in thalattosuchian crocodylomorphs"

_PeerJ, doi:10.7717/peerj.15353_

## Round 0.1 · original submission · Minor Revisions

I congratulate the authors on their fine work, and have no additional concerns to address other than those highlighted by the reviewers.

Reviewer 1 ·

Basic reporting

No comment

Experimental design

No comment

Validity of the findings

No comment

Additional comments

This study provides interesting insights into the unique (among crocodylomorphs) palatal grooves of thalattosuchians. The study is of excellent quality, well-written and easy to understand, with clear and sufficiently detailed figures. The methods are appropriate for this type of study and thoroughly explained in case future researchers wish to replicate the results. The hypothesized implications of these structures make sense given the current data, and the arguments provided by the authors is substantiated. Overall, I find this study to be of great importance to thalattosuchian anatomy and palaeobiology. The quality of the manuscript is of such high degree that I have very few remarks. My remarks, outlined below, are minor and should not take much effort to implement.

My congratulations to the authors on their excellent work!

- In the caption for figure 3, the authors state that the Osteolaemus tetraspis specimen FMNH 98936 is of an adult individual. I have worked with the CT data of the same, and I am not convinced that FMNH 98936 is morphologically mature. There are several morphological features in FMNH 98936 that are more alike immature O. tetraspis than mature O. tetraspis. For one, the supratemporal fenestrae are sub-elliptical and have their long axes oriented anterolaterally (in mature extant crocodylians, including Osteolaemus spp., the supratemporal fenestrae are circular to subcircular and the long axes are aligned anteroposteriorly); the occipital condyle of FMNH 98936 is smaller than the foramen magnum (in mature O. tetraspis the foramen magnum and occipital condyle are equal to sub-equal in width); the cranial ornamentation is not as intense as in adult individuals; the basioccipital plate has a mild posteroventral orientation, whereas in adults it is oriented fully posteriorly; and the brain endocast (based on a personal digital reconstruction) still has prominent impressions of the optic lobes and their surrounding dural envelope (in mature crocodylians, including O. tetraspis, the optic lobes have faint to virtually indistinguishable impressions on the endocast). Therefore, unless the authors have additional data on FMNH 98936 (such as the age of the specimen at the time of death), I believe it may be more accurate to classify it as a subadult instead of an adult. I attach several images to my review where I illustrate the relevant features in FMNH 98936 that make me suspect it is not morphologically mature. I must note that regardless of the true ontogenetic status of FMNH 98936, it has no effect on the conclusions of this study.

- Line 145: please add "are" between "that" and "noticeably" so that the sentence reads "...have palatal canals that are noticeably...".

- Line 241: please remove "seem" before "consider".

- Figure 12 caption: if possible, can the authors please provide the specimen number for the figured humpback whale skull.

Annotated reviews are not available for download in order to protect the identity of reviewers who chose to remain anonymous.

·

Basic reporting

no comment

Experimental design

no comment

Validity of the findings

no comment

Additional comments

Young et al. describe one of the unique morphologies of thalattosuchians – the well-developed elongate grooves present on the secondary palate. They investigate a large number of thalattosuchian and extant crocodylian palates to track this morphology, showing that in thalattosuchians, the grooves extend from canals which connect the nasal and oral cavities, while these grooves and canals are entirely absent in crocodylians. Based on the position of vasculature in extant crocodylians, they propose that the palatal canals (and grooves) transmitted the medial nasal vessels (or novel branches thereof). They also suggest that these canals created a novel pathway for thermoregulation between the palatal plexus and vasculature of the eye and brain in Thalattosuchia. Finally, they show that some other secondarily marine tetrapods – cetaceans and pinnipeds – also have deep vascular grooves on their palates which may have some thermoregulatory function. I think this is a well written manuscript. I have a few suggestions for additional taxa they should discuss in terms of palatal openings, and I would like to see a little more description of the distribution/morphology of the palatal grooves in Thalattosuchia. Otherwise, I think this is an excellent manuscript – another one of those oddities of thalattosuchian morphology I’ve always been interested in, but never got around to systematically investigating myself, so I’m happy to see these authors actually do so. My comments are below and I've also made a number of minor comments on the manuscript pdf.


Comments:

There are three paragraphs in the results section that I think could be largely eliminated (lines 177-225). These paragraphs merely list taxa that lack palatal grooves. As such, I think they could be replaced by a sentence or 2, and the taxon lists moved to tables (e.g., We can find no evidence of palatal grooves in any extant or extinct non-thalattosuchian crocodylomorph based on first-hand examination of specimens (Table X) or from the literature (Table X).

The authors state that the only non-thalattosuchian crocodylomorphs with palatal foramina or grooves are Simosuchus and Lohuecosuchus. However, there are additional taxa/groups I think they should discuss here. Some other notosuchians – namely Notosuchus (Barrios et al., 2018), Mariliasuchus (Zaher et al., 2006), Llanosuchus (Fiorelli et al., 2016), and Caipirasuchus stenognathus (Pol et al., 2014) - have paired openings at the palatine/maxilla suture (referred to as "maxilla-palatine fenestrae" in these publications). Also worthy of discussion would be the anterior palatal fenestra in shartegosuchids – which is a larger single opening, but topologically similar. In both instances, I think these other openings on the palate open directly dorsoventrally and lack canals and grooves, so they are likely not homologous with the structures in thalattosuchians, but I think it is worth discussing the differences between these features in detail. Another potential point to discuss is that shartegosuchids, while not having distinct grooves leading from the palatal opening, are relatively unique among crocodylomorphs in having an ornamented palate – and in extant crocs, ornamentation pits house rich vascular supply (e.g., Seidel, 1979). So, it is possible that shartegosuchids had a highly vascularized palate, and the open connection between the oral and nasal cavities could have allowed for connections between the vessels of the palate and nasal cavity (and by extension, the orbit and braincase) in a similar fashion to thalattosuchian (though presumably for a different function).

The authors introduce that pinnipeds also have deep palatal grooves for the greater palatine artery, and figure the morphology. However, they don’t really discuss the importance of this, or mention it again, only making comparisons between cetaceans and thalattosuchians. Has no one proposed why pinnipeds have enlarged palatal vasculature? Unless this is the case, I think the authors could expand this part of the manuscript a little bit as the appearance of palatal grooves and their elaboration during the evolution of a more marine clade of pinnipeds (phocids) is a second data point to support that this feature is a marine adaptation.

The only other thing I would like to see elaborated upon a little is the description of the presence/morphology of these grooves on the palatines, as this seems to differ between metriorhynchids and other thalattosuchians. In general, I can’t think of any teleosauroid taxa where the grooves extend posteriorly onto the surface of the palatines (i.e., they just extend anteriorly from the canals/foramina onto the maxillae), whereas in metriorhynchids, they often also extend onto the palatines, sometimes terminating as a series of pits just beyond the end of the groove (as in the brachyrhynchus specimen shown in Fig. 1C). In Pelagosaurus , it appears there is a single shallow, U-shaped trough extending onto the palatines posterior to the canals, rather than two distinct grooves (also visible in Fig. 1B). Anyway, I would appreciate a more detailed description of the groove system among Thalattosuchia. The manuscript is pretty short, so I think there is plenty of room to expand on this (especially if the paragraphs listing taxa that lack palatal grooves are moved to tables, as I suggested above).

---

## Round 0.2 · accepted · Accept

The original Academic Editor is not available so I have taken over handling this submission.

Thank you for your close attention to the comments from the reviewers. I have evaluated your revisions, and all seems to be satisfactorily in order. The manuscript is now ready to move forward.